# Association between Initial Serum Cholesterol Levels and Outcomes of Patients Hospitalized after Out-of-Hospital Cardiac Arrest: A Retrospective Multicenter Registry Study

**DOI:** 10.3390/jpm12020233

**Published:** 2022-02-07

**Authors:** Juncheol Lee, Heekyung Lee, Jaehoon Oh, Tae Ho Lim, Hyunggoo Kang, Byuk Sung Ko, Yongil Cho

**Affiliations:** Department of Emergency Medicine, College of Medicine, Hanyang University, Seoul 04763, Korea; doldoly@hanyang.ac.kr (J.L.); massdt@naver.com (H.L.); erthim@gmail.com (T.H.L.); emer0905@gmail.com (H.K.); postwinston@gmail.com (B.S.K.); joeguy@hanmail.net (Y.C.)

**Keywords:** cholesterol, heart arrest, prognosis, cerebral performance category

## Abstract

Purpose: This study aimed to investigate the association between total serum cholesterol levels and outcomes upon discharge in patients after out-of-hospital cardiac arrest (OHCA). Methods: We performed a retrospective observational study using the Korean Cardiac Arrest Resuscitation Consortium (KoCARC) registry. Patients after OHCA whose total serum cholesterol levels were measured within 24 h after arriving at the emergency department were included in the analysis. The association between total serum cholesterol level and neurological outcomes upon discharge and survival to discharge was estimated. Results: Of the 12,321 patients after OHCA enrolled in the registry from October 2015 to June 2020, 689 patients were included. The poor neurologic outcome upon discharge group had a statistically significant lower total serum cholesterol level compared to the good neurologic outcome group (127.5 ± 45.1 mg/dL vs. 155.1 ± 48.9 mg/dL, *p* < 0.001). As a result of multivariate logistic regression analysis, the odds ratio for the neurologic outcome of total serum cholesterol levels was 2.00 (95% confidence interval [CI] 1.01–3.96, *p* = 0.045). The odds ratio for in-hospital death was 1.72 (95% CI 1.15–2.57, *p* = 0.009). Conclusions: Low total serum cholesterol levels could be associated with poor neurologic outcomes upon discharge and in-hospital death of patients hospitalized after OHCA.

## 1. Introduction

An ischemia-reperfusion injury occurs during cardiac arrest and return of spontaneous circulation (ROSC) after cardiopulmonary resuscitation (CPR), and these injuries are comprehensively referred to as post-cardiac arrest syndrome [1].

The pathophysiology of post-cardiac arrest syndrome is complex [2]. During the ischemia period, cellular oxygen storage due to low metabolism is reduced, and prolonged periods cause cellular and tissue damage. Oxidative free radicals are induced by reperfusion injury from chest compressions or ROSC. Oxidative free radicals result in disseminated endothelial injuries and apoptosis, and prolonged injuries lead to a systemic inflammatory response and multiple organ failure [3,4,5,6]. The pathophysiology of ischemia-reperfusion injury is similar to that of sepsis [7].

Hypercholesterolemia is associated with coronary artery disease and stroke. Atherosclerosis occurs when low-density lipoprotein (LDL) particles accumulate in the subendothelial space of the arteries, which causes coronary artery disease and stroke. In addition, untreated dyslipidemia is one of the strongest causes of in-hospital mortality due to coronary artery disease [8].

On the other hand, contrary to the generally accepted theories that low cholesterol levels are good, several studies have shown that low serum cholesterol level is associated with incidence of cancer risk and poor prognosis and mortality of patients of sepsis [9,10,11].

Cholesterol plays a major role in diseases causing critical illness. In general, severe diseases are associated with a catabolic stress state for the systemic inflammatory response [12]. Lipoproteins bind and remove toxic bacterial toxins and endotoxins during the systemic inflammatory response [13,14,15]. Patients with post-ROSC have systemic inflammatory responses similar to those in other patients who are critically ill.

The aim of this study was to assess the association between serum cholesterol levels and outcomes of patients hospitalized after out-of-hospital cardiac arrest (OHCA). We hypothesized that resuscitated patients with initial low serum cholesterol levels would have a poor outcome at survival to discharge.

## 2. Material and Methods

### 2.1. Study Design and Setting

This study was conducted as a multicenter retrospective observational study using the Korean Cardiac Arrest Resuscitation Consortium (KoCARC) registry, a nationwide OHCA registry based on the Utstein templates, and a hospital-based collaborative research network. In addition, it is a data collection system designed to effectively and professionally study evidence for strengthening the survival chain of patients who experience out-of-hospital cardiac arrest [16].

The KoCARC registry included patients who were brought to the participating emergency department by the emergency medical services with resuscitation efforts after OHCA. Patients who experience cardiac arrest of definite non-medical etiology, such as trauma, poisoning, burns, drowning, asphyxia, or hanging, were excluded. Patients with terminal stage illnesses documented in medical records, patients under hospice care, patients with previously documented do-not-resuscitate orders, and pregnant patients were also excluded.

The KoCARC is an organized collaborative research network, which consists of seven research committees, including Epidemiology and Preventive Research, Community Resuscitation Research, EMS Resuscitation Research, Hospital Resuscitation Research, Hypothermia and Post-resuscitation Care Research, Cardiac Care Resuscitation, and Pediatric Resuscitation Research Committees, in which each committee has core and supplemental variables.

The study was approved by the Institutional Review Board (IRB) of the participating hospitals. Informed consent was waived by the IRB due to the retrospective study. The project was registered at the ClinicalTrials.gov (identifier: NCT03222999).

### 2.2. Study Participants

We included patients enrolled in the KoCARC registry from October 2015 to June 2020. The following patients were included in this study: (1) adult patients aged ≥18 years, (2) patients who were hospitalized after OHCA and had successful ROSC at the hospital, (3) patients whose serum total cholesterol levels were measured within 24 h after arriving at the emergency department, and (4) patients with cerebral performance category (CPC) information upon discharge. This study excluded (1) patients aged <18 years, (2) patients who did not survive upon hospital admission, (3) patients who had prehospital ROSC, (4) patients whose serum total cholesterol levels were not measured within 24 h after arriving at the emergency department; and (5) patients without CPC information upon discharge.

### 2.3. Data Collection

The core variables and supplement variables of the KoCARC registry extracted in this study were as follows: (1) patient characteristics: age, sex, hypertension, diabetes, and dyslipidemia; (2) cardiac arrest characteristics: arrest place, witnessed cardiac arrest, bystander CPR, initial rhythm; (3) hospital information: target temperature management (TTM), CPC upon discharge, and survival to discharge; and (4) blood test results: sodium, potassium, albumin, glucose, total cholesterol, lactic acid levels, and arterial blood gas analysis results.

### 2.4. Primary and Secondary Outcomes

In this study, the primary outcome was the neurologic outcome upon discharge of a patient hospitalized after OHCA. In addition, survival to discharge was investigated as a secondary outcome. We also investigated factors that affected neurologic outcomes and mortality.

A good neurologic outcome was defined as CPC 1 and 2, whereas poor neurologic outcomes were defined as CPC 3–5.

### 2.5. Statistics

The data of the KoCARC registry were compiled using a standard spreadsheet application (Excel 2016; Microsoft, Redmond, WA, USA) and were analyzed using R (version 3.6.1, www.R-project.org). Kolmogorov–Smirnov tests were performed for normal distribution in all datasets. Descriptive statistics were used to describe the baseline characteristics of the study. Categorical variables were presented as frequencies and percentages and continuous variables as medians (quartiles) or mean ± standard deviation (SD). An independent *t*-test or Mann-Whitney *U*-test was used to compare continuous variables, and the Chi-square or Fisher’s exact tests were used for categorical variables. Differences were considered statistically significant when the *p*-value was less than 0.05.

We analyzed the area under the receiver operating characteristics curve (AU-ROC) to examine the prognostic performance of total cholesterol level for predicting the neurologic outcome and the survival to discharge. Cut-off values were determined using the Youden’s index with high specificity for predicting poor neurologic outcomes in the ROC curve analysis.

Multivariable logistic regression analysis was performed to determine the association between serum total cholesterol levels and neurologic outcomes or survival to discharge. All variables with *p* values < 0.05 in univariate comparisons were included in the multivariable regression analysis. A variable with a variance inflation factor of >10 was considered multicollinear and was removed from the variable set. We used a backward elimination, sequentially eliminating variables with a threshold of *p* > 0.10, to assess a final adjusted variable set. We used categorical variables based on the cut-off value obtained from Youden’s index to elucidate the type of association between the blood test results and neurologic outcomes for multivariable logistic regression analysis. In addition, age was converted into a categorical variable based on the age of 65 years and the standard of the elderly for logistic regression analysis. Logistic regression analysis results were presented as odds ratios (ORs) and 95% confidence intervals (CIs).

## 3. Results

### 3.1. Baseline Characteristics of OHCA Patients

From 1 October 2015 to 30 June 2020, 12,321 patients were enrolled in the KoCARC registry of the 33 participating hospitals. Except for those who died or were transferred out, 3382 patients survived upon admission after OHCA. We excluded patients who had prehospital ROSC (*n* = 510), had missing data for serum total cholesterol level (*n* = 1596), CPC score (*n* = 229), and other missing data (*n* = 328). Thus, a total of 689 patients were eligible for this study (Figure 1).

The clinical characteristics of the patients in the good neurologic outcome and poor neurologic outcome groups are summarized in Table 1. Of 689 patients who met the inclusion criteria, 81.

Patients showed good neurologic outcomes. Patients with good neurologic outcomes were younger and had a low incidence of diabetes mellitus (DM). Incidences of witnessed cardiac arrest and initial shockable rhythm were significantly higher in the good neurologic outcome group than in the poor neurologic outcome group. Patients with lower potassium, higher albumin, higher pH, lower partial pressure of carbon dioxide (PaCO2), and lower lactate levels were associated with good neurologic outcomes compared to those in patients in the poor neurologic outcome group. In addition, patients with good neurologic outcomes had higher levels of total cholesterol than those with poor neurologic outcomes (127.5 ± 45.1 mg/dL vs. 155.1 ± 48.6 mg/dL, *p* <0.001).

The clinical characteristics stratified by survival to discharge are shown in Table 2. Patients who were alive at discharge had higher levels of total cholesterol than those who died in the hospital (124.0 ± 44.1 mg/dL vs 146.8 ± 47.7 mg/dL, *p* <0.001).

### 3.2. Prognostic Performance of Serum Total Cholesterol Levels for Neurologic Outcomes and Survival to Discharge

The prognostic performance of serum total cholesterol levels for predicting good neurologic outcomes at discharge for patients hospitalized after OHCA was analyzed using the ROC curve (Figure 2). The AUC of total cholesterol was 0.672 (95% CI: 0.613–0.701, *p* < 0.001). The optimal cut-off value of the total cholesterol level for predicting good neurologic outcomes using the Youden’s index was calculated as 119 mg/dL, and for predicting poor neurologic outcomes with high specificity was calculated as 58 mg/dL (Table 3).

In addition, the prognostic performance of serum total cholesterol levels for predicting good neurologic outcomes at discharge for patients hospitalized after OHCA was analyzed using the ROC curve (Figure 3). The AUC of total cholesterol was 0.646 (95% CI: 0.603–0.689, *p* <0.001).

### 3.3. Relationship between Serum Total Cholesterol Levels and Neurologic Outcomes or Survival to Discharge for Patients Hospitalized after OHCA

The following variables showed significant correlation with neurologic outcomes at hospital discharge for patients hospitalized after OHCA in the univariate analysis: age, DM, witnessed cardiac arrest, initial shockable rhythm, and potassium, albumin, total cholesterol, pH, PaCO2, and lactate levels. We performed a multivariate logistic regression analysis of the data of patients hospitalized after OHCA with good neurologic outcomes (Table 4). The factors that associated the neurologic outcomes at hospital discharge for patients hospitalized after OHCA were age, witnessed cardiac arrest, initial shockable rhythm, and albumin, total cholesterol, pH, and PaCO2 levels. The odds ratio for the neurological outcomes and total cholesterol level was 2.00 (95% CI: 1.01–3.96, *p* = 0.045).

In addition, we analyzed the multivariate logistic regression analysis of the data of patients hospitalized after OHCA who survived to discharge (Table 5). The factors that associated the survival to discharge for patients hospitalized after OHCA were DM, initial shockable rhythm, TTM, albumin, total cholesterol, pH, and lactate. The odds ratio for the survival to discharge and total cholesterol level was 1.72 (95% CI 1.15–2.57, *p* = 0.009).

## 4. Discussion

In this study, we performed a multicenter retrospective observational study using the KoCARC registry. We assessed the association between serum cholesterol levels and outcomes of patients hospitalized after OHCA who had successful ROSC at the hospital. We performed a multivariate logistic regression of the data of patients hospitalized after OHCA with good neurologic outcomes, and the serum total cholesterol level was statistically significant as an independent factor (*p* = 0.045). A low initial serum total cholesterol level after ROSC was associated with poor neurological outcomes. In addition, a total cholesterol level higher than 119 mg/dL predicted a good neurologic outcome, and lower than 58 mg/dL predicted a poor neurologic outcome.

In previous studies, Chae et al. reported that a low total cholesterol level upon admission was associated with poor neurologic outcomes in patients with a post cardiac arrest syndrome (PCAS), and could be an easily obtained biomarker for neurologic outcome [17]. In addition, Lee et al. showed that the higher serum levels of total cholesterol and HDL within 1 h after ROSC were associated with good neurologic outcomes of patients with OHCA [18]. The results of this study are similar to those of previous studies. However, this study was meaningful in that it was a nationwide multicenter-based study and the number of samples was relatively large.

In the above studies, the median (interquartile range) of total cholesterol levels in the poor neurologic outcome group were 128 mg/dL (102–153) and 123.0 mg/dL (101.5–157.5), respectively. In this study, mean ± SD the total cholesterol level in the poor neurologic outcome group was 127.5 ± 45.1 mg/dL [17,18]. The definition of the level of hypocholesterolemia differs from each study [19,20]. Total cholesterol levels of patients in the poor outcome group in these studies are either lower or within the normal range according to interquartile range and standard deviation, but an absolute value of prediction for poor outcome cannot be established with these levels.

Most studies have focused on the risk of high cholesterol, such as a risk factor for coronary artery disease [8,21]. However, recent studies have presented roles for cholesterol in critically ill patients. There are several possible causes as to why low cholesterol levels may be associated with a poor neurological prognosis in systemic inflammatory critically ill patients, such as those with post-cardiac arrest syndrome.

First, systemic inflammation leads to an increase in proinflammatory cytokines and endotoxins. Lipoproteins can bind to and remove toxic endotoxins associated with the systemic inflammatory responses. In previous animal studies, transgenic mice with high HDL or LDL concentrations were resistant to the lipopolysaccharide (LPS) challenge [22], where lipoproteins reduced the cytokine response and led to reduced mortality [22,23]. Pajkrt et al. reported that injection of reconstituted HDL reduced tumor necrosis factor, interleukin 6 and 8 secreted by endotoxin in a human double-blind study [24].

In addition, cardiac arrest can lead to reduced total cholesterol levels due to decreased synthetic and secretory functions by liver damage [25]. Hosadurg et al. reported that the out-of-hospital sudden expected death group had lower total cholesterol than the control group [26]. In an animal study, total cholesterol levels before cardiac arrest were higher than those after ROSC [27].

It is important to predict the neurological prognosis of patients with ROSC after cardiac arrest. The 2020 American Heart Association (AHA) guidelines recommend using clinical findings, imaging modalities, and blood biomarkers to predict and evaluate the neurologic prognosis of patients who experience cardiac arrest [28]. Neuron-specific enolase and S-100B are the most commonly examined blood biomarkers for predicting neurologic outcomes. However, these markers are not routinely obtained and have low sensitivity and an inconsistent threshold for predicting neurologic outcomes after ROSC [28]. In this study and a previous study, total cholesterol levels showed potential as a predictor of neurologic outcomes in patients with ROSC after cardiac arrest [17]. Because testing for total cholesterol levels is performed routinely, medical costs can be lower than those of other biomarkers. In contrast, baseline total cholesterol levels prior to cardiac arrest in patients with ROSC vary. Patients may have normal cholesterol levels or have high total cholesterol levels due to hyperlipidemia, or, although the patients have hyperlipidemia, it may be the patients whose total cholesterol level was normal or lower than normal through treatment with statins. For these reasons, total cholesterol level might have little impact as a biomarker for predicting the neurological prognosis in patients who experience cardiac arrest.

However, although baseline total cholesterol levels vary among patients, low cholesterol levels might be a pre-existing risk factor for systemic inflammatory responses such as PCAS or sepsis. Lagrost et al. reported that a pre-existing low level of cholesterol before elective cardiac surgery with cardiopulmonary bypass may be a simple biomarker for the early identification of patients with a high risk of sepsis [29].

This finding showed the potential of low cholesterol itself as a risk factor for severe disease.

This study had several limitations. First, it is a retrospective observational study using a multicenter-based registry. Second, since this study was conducted in one country, it had to be limited by race and nationality. However, the KoCARC registry was used in order to represent South Korea better. Third, the outcome of this study was the evaluation of the neurological outcomes upon discharge and survival to discharge, and the mid- to long-term outcomes were not evaluated. Therefore, further studies are needed to investigate the association between total cholesterol levels and mid- to long-term neurologic and survival outcomes after OHCA. Fourth, of 12,321 OHCA patients, only 689 patients were included. More than 8500 people died, and missing values in blood tests such as total cholesterol were excluded among the surviving hospitalized patients. Because many patients were excluded, an opportunity for selection bias may arise. Fifth, because the cholesterol level is the result of a blood test within 24 h after ROSC, we did not investigate the association between serial changes in total cholesterol levels and outcomes of patients with ROSC. Sixth, because there was not enough information on HDL and LDL in the KoCARC registry, we could not investigate the relationship between detailed lipid profiles and the neurological outcomes and survival to discharge of patients who experience cardiac arrest. Finally, the KoCARC registry does not investigate information about the lipid lowering therapy of patients. So, we could not analyze the association between cholesterol level by lipid lowering therapy and the outcomes in this study.

## 5. Conclusions

Low total serum cholesterol levels within 24 h after arriving at the emergency department of patients after OHCA could be associated with poor neurologic outcomes upon discharge and in-hospital death. In addition, total serum cholesterol might be a biomarker for predicting neurologic outcomes or survival to discharge of hospitalized patients after OHCA.

## Figures and Tables

**Figure 1 jpm-12-00233-f001:**
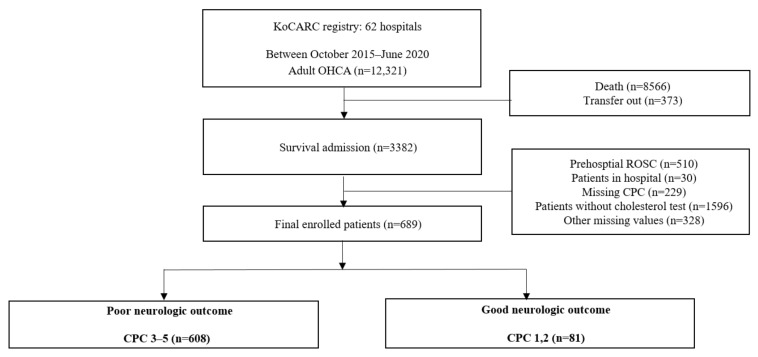
Flow chart of the study (KoCARC, Korean Cardiac Arrest Resuscitation Consortium; OHCA, out-of-hospital cardiac arrest; ROSC, Return of spontaneous circulation; CPC, Cerebral Performance Category).

**Figure 2 jpm-12-00233-f002:**
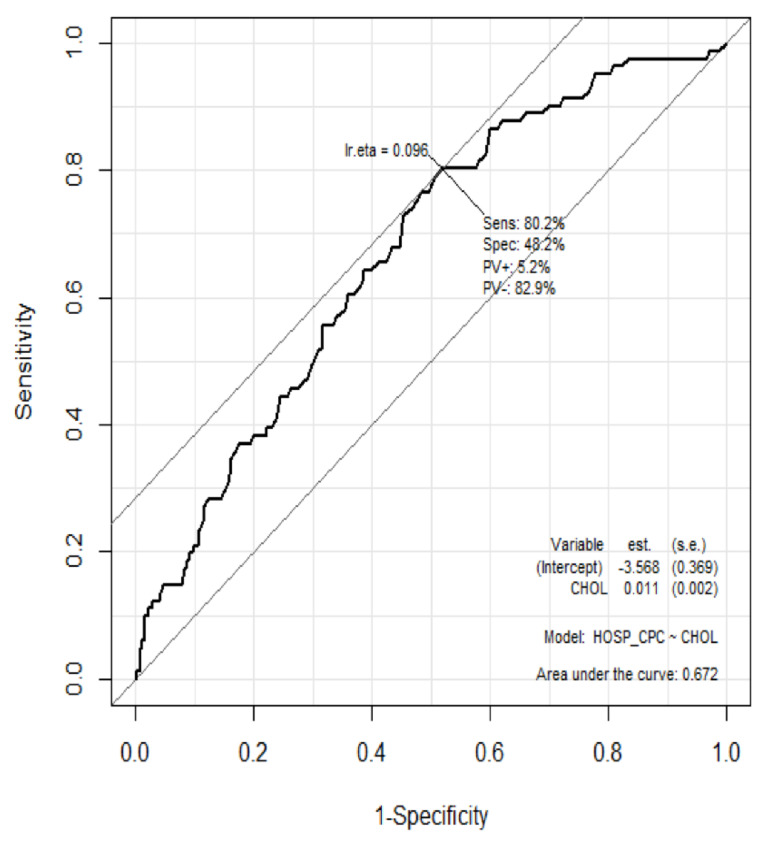
Receiver operator characteristic curve for prediction of neurologic outcomes using total cholesterol levels.

**Figure 3 jpm-12-00233-f003:**
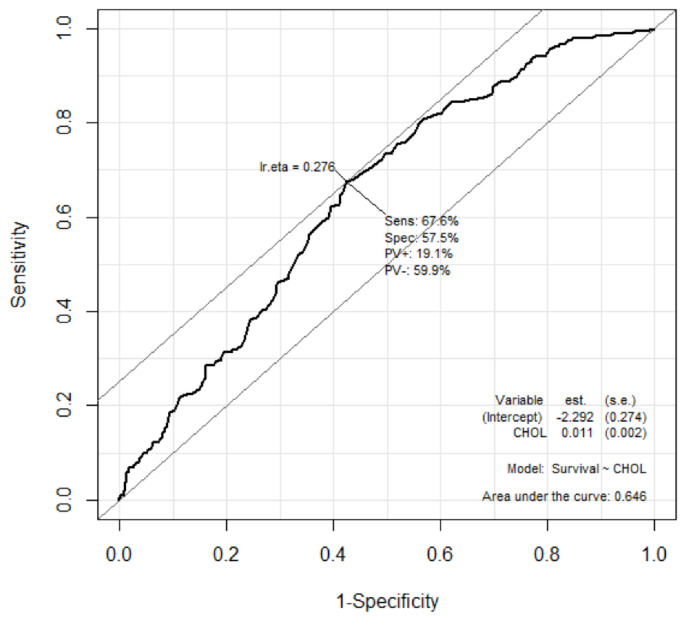
Receiver operator characteristic curve for prediction of survival to discharge using total cholesterol levels. The optimal cut-off value of the total cholesterol level for predicting good neurologic outcomes using the Youden’s index was calculated as 126 mg/dL, and for predicting poor neurologic outcomes with high specificity was calculated as 58 mg/dL (Table 3).

**Table 1 jpm-12-00233-t001:** Baseline characteristics of out-of-hospital cardiac arrest patients with neurologic outcome.

	Poor Neurologic Outcome (*N* = 608)	Good Neurologic Outcome (*N* = 81)	* *p*-Value
Patient characteristic			
Age	64.7 ± 17.2	56.7 ± 16.9	<0.001 *
Gender (male), *n* (%)	428 (70.4)	56 (69.1)	0.918
HTN, *n* (%)	302 (49.7)	39 (48.1)	0.889
DM, *n* (%)	226 (37.2)	15 (18.5)	0.001 *
Dyslipidemia, *n* (%)	96 (15.8)	8 (9.9)	0.218
Cardiac arrest characteristic			
Arrest place			0.001 *
Home/Residence	330 (54.3)	26 (32.1)	
Public place	103 (16.9)	17 (21.0)	
Other	175 (28.8)	38 (46.9)	
Witnessed, *n* (%)	431 (70.9)	72 (88.9)	<0.001 *
Bystander CPR, *n* (%)	265 (43.6)	30 (37.0)	0.318
Shockable, *n* (%)	102 (16.8)	40 (49.4)	<0.001 *
Hospital care			
TTM, *n* (%)	224 (36.8)	29 (35.8%)	0.952
Blood test results			
Na, mEq/L	138.9 ± 7.2	139.8 ± 6.0	0.24
K, mEq/L	5.2 ± 1.6	4.4 ± 1.2	<0.001 *
Albumin, g/dL	3.1 ± 0.7	3.5 ± 0.6	<0.001 *
Glucose, mg/dL	284.0 ± 140.7	276.8 ± 119.6	0.659
Cholesterol, mg/dL	127.5 ± 45.1	155.1 ± 48.6	<0.001 *
pH	6.9 ± 0.2	7.1 ± 0.2	<0.001 *
PaCO2, mmHg	77.1 ± 28.8	56.5 ± 25.7	<0.001 *
PaO2, mmHg	88.2 ± 93.1	94.7 ± 88.1	0.553
Lactate, mmol/L	11.8 ± 4.8	10.4 ± 4.9	0.015 *

Categorical and continuous variables are represented by number (%) and mean ± SD, respectively. The two groups were compared using the independent *t*-test for continuous variables, and chi-square test for categorical variables. HTN; hypertension, DM; diabetes mellitus, CPR; cardiopulmonary resuscitation, * *p* < 0.05 is significant.

**Table 2 jpm-12-00233-t002:** Baseline characteristics of out-of-hospital cardiac arrest patients with survival to discharge.

	Non-Survival (*N* = 485)	Survival (*N* = 204)	*p*-Value
Patient characteristic			
Age	65.1 ± 17.3	60.7 ± 17.2	0.003 *
Gender (male), *n* (%)	339 (69.9)	145 (71.1)	0.827
HTN, *n* (%)	239 (49.3)	102 (50.0)	0.929
DM, *n* (%)	186 (38.4)	55 (27.0)	0.006 *
Dyslipidemia, *n* (%)	80 (16.5)	24 (11.8)	0.142
Cardiac arrest characteristic			
Arrest place			<0.001 *
Home/Residence	275 (56.7)	81 (39.7)	
Public place	81 (16.7)	39 (19.1)	
Other	129 (26.6)	84 (41.2)	
Witness, *n* (%)	340 (70.1)	163 (79.9)	0.011 *
Bystander CPR, *n* (%)	218 (44.9)	77 (37.7)	0.097
Shockable, *n* (%)	77 (15.9)	65 (31.9)	<0.001 *
Hospital care			
TTM, *n* (%)	157 (32.4)	96 (47.1)	<0.001 *
Blood test results			
Na, mEq/L	138.8 ± 7.6	139.5 ± 5.6	0.184
K, mEq/L	5.2 ± 1.6	4.6 ± 1.3	<0.001 *
Albumin, g/dL	3.0 ± 0.7	3.4 ± 0.6	<0.001 *
Glucose, mg/dL	281.1 ± 142.3	288.1 ± 128.9	0.545
Cholesterol, mg/dL	124.0 ± 44.1	146.8 ± 47.7	<0.001 *
pH	6.9 ± 0.2	7.0 ± 0.3	<0.001 *
PaCO2, mmHg	75.8 ± 28.9	72.1 ± 29.9	0.135
PaO2, mmHg	86.7 ± 89.7	94.3 ± 98.8	0.325
Lactate, mmol/L	12.0 ± 5.0	10.6 ± 4.4	<0.001 *

Categorical and continuous variables are represented by number (%) and mean ± SD, respectively. The two groups were compared using the independent *t*-test for continuous variables, and chi-square test for categorical variables. HTN; hypertension, DM; diabetes mellitus, TTM; target temperature management, CPR; cardiopulmonary resuscitation, * *p* < 0.05 is significant.

**Table 3 jpm-12-00233-t003:** Cut-off and diagnostic values of cholesterol for predicting neurologic outcome and survival.

	Cut-Off Value	Accuracy	Sensitivity	Specificity	PPV	NPV
**Neurologic outcome**						
Cholesterol for predicting poor neurologic outcome	<58.0 mg/dL	48.96%	2.96%	98.7%	94.1%	11.9%
Cholesterol for predicting good neurologic outcome	≧119.0 mg/dL	59.31%	80.2%	48.2%	5.2%	82.9%
**Survival**						
Cholesterol for predicting in hospital death	<58.0 mg/dL	45.22%	3.71%	99.5%	94.7%	30.3%
Cholesterol for predicting survival to discharge	≧126.0 mg/dL	61.29%	67.6%	57.5%	19.1%	59.9%

The cut-off value for predicting poor outcomes was calculated as high specificity, and the cut-off value of cholesterol for predicting good outcomes is calculated as Youden’s index. PPV, positive predictive value; NPV, negative predictive value.

**Table 4 jpm-12-00233-t004:** Multivariate logistic regression of out-of-hospital cardiac arrest patients with neurologic outcome.

	Crude OR (95% CI)	** p*-Value	Adjusted OR (95% CI)	* *p*-Value
Age	0.21 (0.16–0.28)	<0.001 *	0.04 (0.01–0.13)	<0.001 *
Witnessed	3.29 (1.61–6.71)	0.001 *	3.27 (1.48–7.26)	0.003 *
Initial shockable rhythm	4.84 (2.98–7.86)	<0.001 *	3.40 (1.94–5.98)	<0.001 *
Albumin	3.90 (2.31–6.58)	<0.001 *	2.06 (1.11–3.82)	0.022 *
Cholesterol	3.63 (2.05–6.42)	<0.001 *	2.00 (1.01–3.96)	0.045 *
pH	4.71 (2.83–7.85)	<0.001 *	2.19 (1.16–4.12)	0.015 *
PaCO2	0.18 (0.11–0.3)	<0.001 *	0.27 (0.14–0.49)	<0.001 *
Lactate	0.37 (0.22–0.62)	<0.001 *	0.56 (0.30–1.05)	0.07 *

* *p* < 0.05 is significant. OR; odds ratio, CI; confidence interval.

**Table 5 jpm-12-00233-t005:** Multivariate logistic regression of out-of-hospital cardiac arrest patients with survival to discharge.

	Crude OR (95% CI)	* *p*-Value	Adjusted OR (95% CI)	* *p*-Value
DM	0.59 (0.41–0.61)	0.004 *	0.66 (0.45–0.98)	0.038 *
Witnessed	1.70 (1.14–2.51)	0.009 *	1.51 (0.99–2.31)	0.056
Initial shockable rhythm	2.48 (1.69–3.63)	<0.001 *	1.56 (1.01–2.39)	0.043 *
TTM	1.86 (1.33–2.59)	<0.001 *	1.92 (1.33–2.76)	<0.001 *
K	0.48 (0.34–0.67)	<0.001 *	0.71 (0.49–1.03)	0.074
Albumin	3.06 (2.18–4.31)	<0.001 *	2.24 (1.53–3.29)	<0.001 *
Cholesterol	2.74 (1.92–3.91)	<0.001 *	1.72 (1.15–2.57)	0.009 *
pH	2.00 (1.43–2.79)	<0.001 *	1.60 (1.09–2.34)	0.016 *
Lactate	0.54 (0.36–0.81)	0.003*	0.66 (0.41–1.05)	0.081

* *p* < 0.05 is significant. OR; odds ratio, CI; confidence interval, DM; diabetes mellitus, TTM; target temperature management.

## Data Availability

Not applicable.

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
