# Peer review of "Association between Initial Serum Cholesterol Levels and Outcomes of Patients Hospitalized after Out-of-Hospital Cardiac Arrest: A Retrospective Multicenter Registry Study"

_jpm, 2022, doi:10.3390/jpm12020233_

Round 1

Reviewer 1 Report

In the article entitled "Association between initial serum cholesterol levels and outcomes of patients hospitalized after out-of-hospital cardiac arrest: a retrospective multicenter registry study” Dr. Lee and colleagues report about serum cholesterol and neurologic outcomes in patients survived to out of hospital cardiac arrest. Data were taken from a large multicenter Korean registry (KoCARC). Despite an initial population of more than 12000 patients, only 689 were eligible for this analysis. Remarkably, the vast majority (88%) had a poor neurologic outcome. Predictors were age, diabetes mellitus, witnessed cardiac arrest and initial shockable rhythm. Looking at the blood samples lower potassium, higher albumin, higher pH, lower partial pressure of carbon dioxide and lower lactate levels were associated with good neurologic outcomes. All these data have already been widely presented and their influence on neurological outcome is not surprising. Higher total cholesterol at admission was correlated with better neurologic outcomes. Unfortunately, LDL and HDL cholesterol levels were not collected for every patient. Authors were able to find cut-off values of cholesterol for predicting neurologic outcome and survival. The several limitations of the study are correctly reported in the manuscript. The conclusions seem to be interesting and the argumentation reasonable and supported by previous works.

Author Response

Thank you for your detailed and rigorous comments. We agreed the opinions about our study and manuscript and tried to revise it according to them.

Sincerely.

Best regard.

Answers for Specific points

Point 1

Only 689 patients qualified for analysis out of 12,321. Please discuss the opportunity for bias – for example, we know nothing about the cholesterol levels of the patients who died prior to admission; perhaps they all had high cholesterol. Please be careful to list your inclusion criteria when you make your inferences as the generalizability of the results is not established.

Answer

Thank you for your kind comment. We agree your opinion. Excluding 8,566 patients who died, there were 3,382 survived hospitalization. Among them, 689 patients were included in this study, and there were many missing values because the total cholesterol test was not performed on all patients. As your comment, bias may occur in this case, so the limitations are additionally described. Thank you again.

Before

After revision

Fourth, although total cholesterol level was a core variable in the KoCARC registry, there were many missing values depending on the hospitals

Fourth, of 12,321 OHCA patients, only 689 patients were included. 8,566 people died, and missing values in blood tests such as total cholesterol were excluded among the surviving hospitalized patients. Because many patients were excluded, an opportunity for selection bias may arise.

Point 2

These data were recorded in a spread of 5 years – guidelines/standards of care and drugs may have changed in this time (pcsk9 availability, for example). Please discuss.

Answer

Thank you for your exact comment. We agree your opinion. PCSK9 inhibitor, a nonstatin drug, was included in the guideline a few years ago and is widely used in other countries. However, in South Korea, PCSK9 inhibitor is only covered by insurance for patients with familial hypercholesterolemia, and only a small number of patients are taking it, so it is expected that there will be little change in total cholesterol levels for dyslipidemia patients according to the guidelines. Thank you again.

Point 3

“several studies have shown that low serum cholesterol level is associated with poor neurological prognosis and mortality in patients who are critically ill such as those with cancer and sepsis [9 – 11]” –reference #9 does not support this sentence; reference 9 examines the association between cholesterol levels and incidence of cancer. Please double check references and update as needed.

Answer

Thank you for your kind comment. We misunderstood about the reference #9.  We revised this part.

Before

After revision

several studies have shown that low serum cholesterol level is associated with poor neurological prognosis and mortality in patients who are critically ill such as those with cancer and sepsis [9 – 11]

several studies have shown that low serum cholesterol level is associated with incidence of cancer risk and poor prognosis and mortality of patients of sepsis [9 – 11]

Point 4

A cutoff yielding a sensitivity of 2.96% (or 3.71%) is not reasonable or useful; it does not make sense to maximize specificity at such great expense of sensitivity. Further, it also does not make sense to model both levels of a binary outcome – please remove the results for models of poor outcomes and death. If a value exceeding the optimal cutoff anticipates a positive outcome, then a value less than the optimal cutoff anticipates a negative outcome. If you are not satisfied with the cutoff suggested by Youden’s optimization, you can consider another optimization method (e.g., point closest to (0,1) on the unit square). Please also provide the overall accuracy according to the cutoffs.

Answer

Thank you for your comment. As a biomarker role for predicting poor neurologic outcome in post-ROSC patients, the cutoff value at the highest specificity was calculated like referenced study. (Chae, M.K.; Lee, S.E.; Min, Y.G.; Park, E.J. Initial Serum Cholesterol Level as a Potential Marker for Post Cardiac Arrest Patient Outcomes. Resuscitation 2020, 146, 50–55, doi:10.1016/j.resuscitation.2019.11.003.). However, as your comment, if you think the sensitivity is too small and inappropriate, we will delete that part. And, ‘accuracy’ will be presented in the table.

Before

After revision

Point 5

Due to the retrospective observational nature of this study, please avoid the strong wording choices such as ‘independently influenced’ – you are unable to make claims of causality with this analysis. Please use ‘associated’ instead.

Answer

Thank you for your kind comment. We agree your opinion. We added in the Discussion section as your comments.

Before

After revision

- The factors that independently influenced the neurologic outcomes at hospital discharge for patients hospitalized after OHCA were age

- The factors that independently influenced the survival to discharge for patients hospitalized after OHCA were DM

- The factors that associated the neurologic outcomes at hospital discharge for patients hospitalized after OHCA were age

- The factors that associated the survival to discharge for patients hospitalized after OHCA were DM

Point 6

Please discuss the observed total cholesterol levels observed in this study and the other referenced studies in the context of dyslipidemia/guidelines. Although the patients who had better outcomes had higher cholesterol than the patients who had worse outcomes, the readers need to know whether or not the observed cholesterol levels in this study were in a normal range.

Answer

Thank you for your kind comment. I agree your opinion. We added in the Discussion section as your comments.

Before

After revision

In the above studies, the median (interquartile range) of total cholesterol levels in the poor neurologic outcome group were 128 mg/dL (102-153) and 123.0 mg/dL (101.5-157.5), respectively. In this study, the total cholesterol level in the poor neurologic outcome group was 127.5 ± 45.1 mg/dL [17,18]. The definition of the level of hypocholesterolemia differs from each study [19,20]. Total cholesterol levels of patients in the poor outcome group in these studies are either lower or within the normal range according to interquartile range and standard deviation, but an absolute value of prediction for poor outcome cannot be established with these levels.

Point 7

Please provide information regarding lipid lowering therapy use in this cohort.

Answer

Thank you for your exact comment. We agree your opinion. However, the KoCARC registry we used does not have information on whether or not patients have been treated with lipid lowering therapy. As your comments, the lack of information on lipid lowering therapy is a limitation. This part was added on the limitation.

Before

After revision

Finally, the KoCARC registry does not investigate information about lipid lowering therapy of patients. So, we could not analyze the association between cholesterol level by lipid lowering therapy and the outcomes in this study.

Point 8

Please be consistent with the order of presentation of groups (for example, the figure lists ‘good’ before ‘poor,’ while the table lists ‘poor’ before ‘good’).

Answer

We revised the figure 1. Thank you for your kind comment.

Before

After revision

Point 9

Please change ‘Reterun’ to ‘Return’ in the caption of Figure 1.

Answer

Thank you for your kind comment. We changed ‘Reterun’ to ‘Return’ in the caption of Figure 1.

Before

After revision

Reterun of spontaneous circulation

Return of spontaneous circulation

Point 10

Please double check the formatting of subheadings throughout (for example, “Prognostic performance of serum total cholesterol levels for neurologic outcomes and survival to discharge”)

Answer

Thank you for your comment. We changed the subheadings format

Before

After revision

-Prognostic performance of serum total cholesterol levels for neurologic outcomes and survival to discharge

-Relationship between serum total cholesterol levels and neurologic outcomes or survival to discharge for patients hospitalized after OHCA

3.1. Baseline characteristics of OHCA patients

3.2. Prognostic performance of serum total cholesterol levels for neurologic outcomes and survival to discharge

3.3. Relationship between serum total cholesterol levels and neurologic outcomes or survival to discharge for patients hospitalized after OHCA

Point 11

Please change “First, systemic inflammation lead to increase the proinflammatory cytokine and endotoxin.” To “First, systemic inflammation leads to increases in proinflammatory cytokines and endotoxins.”

Answer

Thank you for comment. We changed the word as your comment.

Before

After revision

First, systemic inflammation lead to increase the proinflammatory cytokine and endotoxin

First, systemic inflammation leads to increases in proinflammatory cytokines and endotoxins

Reviewer 2 Report

Summary: The authors analyzed data from 689 patients who had an out-of-hospital cardiac arrest and had cholesterol levels measured within 24 hours of ED. The authors found that patients with ‘good neurologic outcomes’ had higher serum cholesterol levels. The authors conclude that total cholesterol can be used to predict neurologic outcomes and hospital survival.

Major Points:

Only 689 patients qualified for analysis out of 12,321. Please discuss the opportunity for bias – for example, we know nothing about the cholesterol levels of the patients who died prior to admission; perhaps they all had high cholesterol. Please be careful to list your inclusion criteria when you make your inferences as the generalizability of the results is not established.

These data were recorded in a spread of 5 years – guidelines/standards of care and drugs may have changed in this time (pcsk9 availability, for example). Please disucss.

“several studies have shown that low serum cholesterol level is associated with poor neurological prognosis and mortality in patients who are critically ill such as those with cancer and sepsis [9 – 11]” –reference #9 does not support this sentence; reference 9 examines the association between cholesterol levels and incidence of cancer. Please double check references and update as needed.

A cutoff yielding a sensitivity of 2.96% (or 3.71%) is not reasonable or useful; it does not make sense to maximize specificity at such great expense of sensitivity. Further, it also does not make sense to model both levels of a binary outcome – please remove the results for models of poor outcomes and death. If a value exceeding the optimal cutoff anticipates a positive outcome, then a value less than the optimal cutoff anticipates a negative outcome. If you are not satisfied with the cutoff suggested by Youden’s optimization, you can consider another optimization method (e.g., point closest to (0,1) on the unit square). Please also provide the overall accuracy according to the cutoffs.

Due to the retrospective observational nature of this study, please avoid the strong wording choices such as ‘independently influenced’ – you are unable to make claims of causality with this analysis. Please use ‘associated’ instead.

Please discuss the observed total cholesterol levels observed in this study and the other referenced studies in the context of dyslipidemia/guidelines. Although the patients who had better outcomes had higher cholesterol than the patients who had worse outcomes, the readers need to know whether or not the observed cholesterol levels in this study were in a normal range.

Please provide information regarding lipid lowering therapy use in this cohort.

Minor Points:

Please be consistent with the order of presentation of groups (for example, the figure lists ‘good’ before ‘poor,’ while the table lists ‘poor’ before ‘good’).

Please change ‘Reterun’ to ‘Return’ in the caption of Figure 1.

Please double check the formatting of subheadings throughout (for example, “Prognostic performance of serum total cholesterol levels for neurologic outcomes and survival to discharge”)

Please change “First, systemic inflammation lead to increase the proinflammatory cytokine and endotoxin.” To “First, systemic inflammation leads to increases in proinflammatory cytokines and endotoxins.”

Author Response

Thank you for your detailed and rigorous comments. We considered again for all of your comments as follows. Sincerely.

Best regard.

Answers for Specific points

Point

In the article entitled "Association between initial serum cholesterol levels and outcomes of patients hospitalized after out-of-hospital cardiac arrest: a retrospective multicenter registry study” Dr. Lee and colleagues report about serum cholesterol and neurologic outcomes in patients survived to out of hospital cardiac arrest. Data were taken from a large multicenter Korean registry (KoCARC). Despite an initial population of more than 12000 patients, only 689 were eligible for this analysis. Remarkably, the vast majority (88%) had a poor neurologic outcome. Predictors were age, diabetes mellitus, witnessed cardiac arrest and initial shockable rhythm. Looking at the blood samples lower potassium, higher albumin, higher pH, lower partial pressure of carbon dioxide and lower lactate levels were associated with good neurologic outcomes. All these data have already been widely presented and their influence on neurological outcome is not surprising. Higher total cholesterol at admission was correlated with better neurologic outcomes. Unfortunately, LDL and HDL cholesterol levels were not collected for every patient. Authors were able to find cut-off values of cholesterol for predicting neurologic outcome and survival. The several limitations of the study are correctly reported in the manuscript. The conclusions seem to be interesting and the argumentation reasonable and supported by previous works.

Answer

Thank you for your kind comments. There were many missing values because the total cholesterol test was not performed on all patients. As your comment, bias may occur in this case, so the limitations are additionally described. And, as your comment, it is unfortunate that there was no detailed information on lipoproteins. Thank you again.

Before

After revision

Fourth, although total cholesterol level was a core variable in the KoCARC registry, there were many missing values depending on the hospitals

Fourth, of 12,321 OHCA patients, only 689 patients were included. 8,566 people died, and missing values in blood tests such as total cholesterol were excluded among the surviving hospitalized patients. Because many patients were excluded, an opportunity for selection bias may arise.